# AmbiTeam: Providing Team Awareness Through Ambient Displays

Sarah Morrison-Smith *
Barnard College

Lydia B. Chilton †
Columbia University

Jaime Ruiz ‡
University of Florida

## ABSTRACT

Due to the COVID-19 pandemic, research is increasingly conducted remotely without the benefit of informal interactions that help maintain awareness of each collaborator's work progress. We developed AmbiTeam, an ambient display that shows activity related to the files of a team project, to help collaborations preserve a sense of the team's involvement while working remotely. We found that using AmbiTeam did have a quantifiable effect on researchers' perceptions of their collaborators' project prioritization. We also found that the use of the system motivated researchers to work on their collaborative projects. This effect is known as "the motivational presences of others," one of the key challenges that make distance work difficult. We discuss how ambient displays can support remote collaborative work by recreating the motivational presence of others.

**Keywords:** Collaboration; remote work; awareness; ambient display

**Index Terms:** Human-centered computing—Human-Computer Interaction—Empirical studies in HCI—

## 1 INTRODUCTION

With the advent of the COVID-19 pandemic, research is increasingly conducted remotely without the affordances of informal interactions that enhance fluidity and interactivity in teams. With the increase in use of collaboration tools for research during the pandemic, developing and improving technology for collaboration is of the utmost importance [33]. Remote collaboration has always faced numerous challenges, such as decreased awareness of colleagues and their context [43] and limited motivational sense of the presence of others [43]. Awareness of one's collaborators is necessary for ensuring that each teammate's contributions are compatible with the collaboration's collective activity [19]. It also plays an essential role in determining whether an individual's actions mesh with the group's goals and progress [19]. The motivational sense of the presence of others complements awareness by producing "social facilitation" effects, like driving people to work more when they are not alone [43].

Similarly, a researcher's perception of their collaborator's effort in a project can profoundly impact collaboration [15]. In particular, researchers tend to feel anxious about the success of their collaboration when they are concerned that competing priorities result in less commitment to the project [15]. The shift to remote work likely exacerbates this challenge since remote researchers lack the awareness of their collaborators' activities.

Together, these challenges pose a significant challenge to collaboration. It is essential that we address these challenges, given that the efficacy of science significantly improves when researchers from diverse backgrounds collaborate on a project [14]. We hypothesize that since a heightened awareness of a collaborator's research activities might reveal project prioritization, improved awareness could

---

*e-mail: smorriso@barnard.edu

†e-mail: chilton@cs.columbia.edu

‡e-mail: jaime.ruiz@ufl.edu

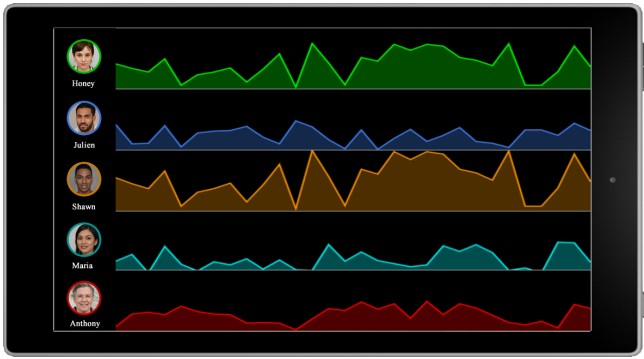

Figure 1: Example visualization of a team's work-related activity which was featured on a tablet with an ambient display in each of our user's workplaces. The visualization shows activity from five fictional teammates using randomly generated data. Each team member has an area graph where each point represents their activity for that day.

lessen the anxiety caused by uncertainty regarding a collaborator's investment. While various existing systems improve awareness in remote teams [9, 10, 23, 24, 35, 38, 44], no solution exists that solves the challenge of perceived prioritization.

To this end, we developed a system, AmbiTeam (shown in Figure 1) to improve a researcher's awareness of their collaborator's project-related activity. The system tracks and visualizes file changes in user-specified project directories to indicate how much effort or work a collaborator has put in on the project. We performed a user evaluation of the system with ten researchers in co-located and remote collaborations to investigate the effect of ambiently providing project-related activity information on a researcher's work behavior and perception of effort. We found that AmbiTeam had some impact on a researcher's motivation to work on the project as well as perceptions of their collaborators' effort. The key contributions of this paper are:

- Increased understanding of how to facilitate team awareness

- A deeper understanding of the motivating effect of awareness on work behavior

- New insights into the impact of increased awareness on perceptions of remote collaborators' effort

## 2 PRIOR WORK

We examine studies on awareness-based systems for supporting collaboration as well as existing solutions for unobtrusively providing information via ambient displays.

### 2.1 Awareness-Based Systems

Several technologies were developed to help remote workers become aware of their collaborator's research activities. Some early technology (e.g., [9, 23, 38]) featured permanently open audiovisual connections between locations, with the idea that providing unrestricted face-to-face communication would enable collaborative work as if the researchers were in the same room. Many modern systems

utilize notification systems to provide awareness of collaborators' activities [36], which is generally considered disruptive [2]. Given the importance of reducing "dramatic changes in work habits" [42], it is likely that an effective system needs to be as unobtrusive as possible. We therefore focus our attention to prior work that utilizes unobtrusive methods (e.g., ambient systems) to promote awareness in collaborative groups.

Several systems have been developed to facilitate awareness of the presence of remote collaborators via mechanisms such as pointer icons [25], virtual shadows of hands [52] and full body sihlouettes [45], simple video textures [26] and avatars [12]. These systems exist to inform collaborators who is currently working on which aspect of the project [12, 25, 26, 52] or facilitate coordination when synchronously collaborating [45].

Additional technologies have focused on providing auxilerary information about a project on public displays [11, 47]. Boden et al. used this method to share small chunks of project-related information that hinted at company activities and indicated which collaborators could be asked for specific questions [11]. Similarly Schwarzer et al. used this approach to provide SCRUM data, such as JIRA team charts and announcements from software architexts as well as build summaries, reports, and errors, to agile projects [47]. Although details regarding collaborator presence and project-specific information can greatly benefit a collaboration, it does not give an indication of effort, which AmbiTeam was designed to provide.

Prior work includes many systems that, like AmbiTeam, provide information regarding current activity (e.g., which files are artifacts users are working on [8, 12], types of current activity [12] and work progress [6]). These technologies provide useful context to collaborative activities involving shared files and; however, because information is limited to current activity, the information is less useful in asynchronous collaborations. AmbiTeam, in contrast, visualizes bot current and past activity, which is useful for both synchronous and asynchronous collaborative activities.

Two systems other than AmbiTeam have been created to track and visualize both current and past work activity. Begole et al. visualized users patterns of work by tracking their mouse clicks, keyboard strikes, location, and when they opened emails. Their goal was to help people determine the best times to interrupt their collaborators [5]. However, unlike AmbiTeam, this visualization not distinguish between project and non-project work, making it difficult to discern project-related effort. In addition, recently, Glikson et al. [24] created a tool that visualizes effort, which is determined by measuring the number of keystrokes that members of a collaboration make in a task collaboration space while playing League of Legends. They found that this tool improved both team effort and performance [24]. However, while we believe that tracking keystrokes can give important insights when playing keyboard-based games such as League of Legends, the relationship between keystrokes and scientific activities, such as generating visualizations of data, is unclear. We therefore used the modified approach of tracking file changes that, as described in Section 3, captures a variety of tasks identified by prior work as being part of the research workflow in the context of collaborative scientific research.

## 2.2 Ambient Displays

Ambient displays are information sources designed to communicate contextual or background information in the periphery of the user's awareness and only require the user's attention when it is appropriate or desired [27]. Mankoff et al.'s heuristics developed to evaluate ambient displays highlights the key characteristics that make a display ambient [37]:

- Provision of useful and relevant information

- "Peripherality" or unobtrusiveness

- Match between its design and the environment

- Conveyance of "just enough" information

- Low cognitive load through consistent and intuitive mapping

- Aesthetic and pleasing design

And, in the case where an ambient display has multiple states or levels of information, the display should also have:

- Fluid transitions between levels of information

- Noticeable state of system

Methods for conveying information via ambient displays include the use of light levels [16, 30], wind [39], temperature [54], music [4], and art [27]. For example, one of the earliest ambient systems, "ambientRoom", used visual displays of water ripples to convey information about the activities of a laboratory hamster and light patches to indicate the amount of human movement in an atrium [30]. Ambient displays are not limited to immersive environments and can also take the form of standalone media displays that allow multiple people to simultaneously receive information [16]. Applications of ambient displays include educating users about resource (e.g., water [32, 35] and power [28]) consumption, improving driving [17, 46], monitoring finances [49], and assisting time management during meetings [41].

Some ambient systems have been developed to support collaboration by tackling the issues of determining availability [1, 13]. One system, "Nimio," used a series of physical toys to indicate the presence and availability of collaborators in separate offices [13]. Toys in one office would cause associated toys in other offices to light up with colored lights when they detected sound and movement, indicating that a collaborator was in their office and communicating whether the collaborator appeared to be busy. Alavi and Dillenbourg [1] placed colored light boxes on tables in a student space that allowed students to indicate their presence, availability, and the coursework they were currently working on so that any given student could be aware of other students with whom they could collaborate.

Streng et al. [50] used ambient displays to convey information about the quality of collaboration between students working on a group task. In this paper, collaboration performance was measured by evaluating student adherence to a collaboration script that specified different phases and tasks to be carried out by individual team members. Performance information was communicated to the student participants either via a diagram featuring charts and numbers or an ambient art display showing a nature scene featuring trees, the sun or moon, and sometimes clouds and rain.

## 2.3 Research Questions and Study Goals

We hypothesize that promoting awareness by providing up-to-date information about a collaborator's project activities will affect a researcher's perception of their collaborator's effort. To avoid dramatically changing work habits, we pursue an ambient-based approach where information is conveyed without requiring the user's attention. To pursue these goals, we sought to answer the following questions:

**RQ1.** Can tracking file activity give teammates a sense of their teammates' efforts?

**RQ2.** Will ambient information on team project activities affect perceptions of collaborators' effort?

**RQ3.** What effects will providing team project activity information have on work behavior?

## 3 SYSTEM DESIGN

### 3.1 Privacy and Scope

Project effort is difficult to characterize as it includes activities that are impossible to track (e.g. thinking about a project) or are potentially sensitive (e.g. emails, phone calls). In order to respect the privacy of users, we avoid monitoring activities such as phone calls and emails and instead focus on the activity of files in user-specified project directories. This allows AmbiTeam to observe project activities related to the various stages of the research life-cycle identified by prior work [40]. For example, during experimentation, the system will be able to detect changes in electronic lab notebooks and cheat sheets used by researchers [40] as well as data. AmbiTeam will also observe data analysis by tracking changes in analysis code or scripts (also discussed in [40]) as well as generated output. Furthermore, the system will be able to monitor publication preparation by detecting changes in writing-related materials.

### 3.2 Activity Tracking

Activity is detected using a desktop application that monitors specified directories for file creation, deletion, and change events. AmbiTeam first prompts the user to select a directory to be watched, and on the back end, monitors the meta-data of the directory's files without viewing the file's contents. Once a file or directory in the watched directory is created, deleted, or changed, the user's ID and the time of the file event is encrypted and sent to a server. This approach is similar to tracking changes made to files by a software developer using source control such as Git or Subversion, except that instead of the user manually logging changes through commits, AmbiTeam automatically tracks and logs all file changes. In the same way that commit activity in collaborative projects has been used to monitor activity in collaborative software development [18], we seek to use file activity to monitory activity in collaborative projects.

### 3.3 Displaying Activity

The number of activities occurring each day for each user is visualized in the form of a point on an area graph. An area graph for each collaborator is displayed on a tablet, showing each day's cumulative activity in real time. The height of the graph on each day indicates the total amount of activity at that time and the area of the graph shows the total amount of activity over the course of a two week window. Activity is normalized across the team to facilitate comparisons between team members. Figure 1 shows an example. Although evaluation of AmbiTeam's display using the heuristics outlined by Mankoff et al. [37] is a topic for future work, AmbiTeam was designed to possess the characteristics of an ambient display outlined in Section 2.2.

## 4 METHOD

### 4.1 Participants

To determine whether AmbiTeam facilitates team awareness, we recruited 10 scientists who are part of four existing collaborations across four institutions in the United States aged 21 to 33 ($\mu = 27.3$, $\sigma = 3.5$, three females). Each of the collaborations is labeled A-D. The research area, title, and group of each participant is presented in Table 1. Participants were recruited inter-departmental email and our methodology was approved by our institutional review board. The configuration of the teams participating in this study ranged from fully remote (team A) to fully co-located (teams C and D). Team B had a mixed composition where participants B2 and B3 were co-located while B1 and B4 were each at different locations. All co-located teams worked in the same offices as their collaborators and reported working closely together.

Our participants sought to answer a variety of scientific questions, which can be broadly summarized as:

Table 1: Participant backgrounds.

| ID | Research Area | Title |
|----|---------------|-------|
| A1 | Biological Anthropology | Post-Doc |
| A2 | Vertebrate Paleontology | Ph.D. Student |
| B1 | Computer Vision and Machine Learning | Master's Student |
| B2 | Computational Linguistics | Post-Doc |
| B3 | Computer Vision and Human-Computer Interaction | Master's Student |
| B4 | Human-Computer Interaction | Ph.D. Student |
| C1 | CyberSecurity | Ph.D. Student |
| C2 | CyberSecurity | Ph.D. Student |
| D1 | CyberSecurity | Ph.D. Student |
| D2 | CyberSecurity | Ph.D. Student |

- **Understanding Faunal Change:** identifying what happens to animals during the major climate events called the paleocene-eocene thermal maximum. (Team A).

- **Enable Communicative Mechanisms Between Humans and Computers:** bringing together human's natural language capability and computers' data processing capability to allow peer-to-peer collaboration between humans and computers. (Team B).

- **Personalized Computer Security:** using personal information to accomplish security tasks like authentication. This includes extracting nuanced personal information (e.g., vocal characteristics) from easily obtained information, such as pictures of people's faces. (Teams C & D).

### 4.2 Procedure

Participants were each given a tablet with AmbiTeam's display, had the activity monitor installed on their work computers, and were instructed on how both the activity monitor and the visualization worked. Participants then completed a pre-test where they estimated the amount of effort that each participating researcher is putting into the project, including themselves, on a scale from 1 to 9 with 1 being "very low" and 9 being "very high." Participants were also asked to explain the reasoning behind their rankings. Over the course of four weeks, on two randomly chosen days a week, participants were asked to repeat this assessment via email. During this time, AmbiTeam's visualization was turned off in order to prevent participants from consulting the visualization, since the goal was to determine whether the system's use affected their perception, not whether they could read the chart. To minimize visualization downtime, participants were given up to 24 hours to respond with their assessment.

At the end of the study, we conducted semi-structured interviews with the participants. By using the semi-structured interview technique, we were able to cover additional topics as they were encountered, reducing the likelihood that important issues were overlooked [34]. When possible, interviews took place at each of the participant's primary workspaces (offices or labs). Participants located at remote locations participated in the interviews over Zoom [29]. Interviews were approximately 30 minutes in duration and were recorded in audio format, then transcribed.

Participants were first asked to educate us about the collaborative research that they participated in during the study including their roles on the project(s) and the goal(s) of the research. We then asked participants to discuss their experiences using AmbiTeam as well as any changes they would propose and their likelihood of using the system in the future.

### 4.3 Qualitative Data Analysis

We performed a bottom-up analysis of participants' responses by constructing an affinity diagram (a.k.a. the KJ method) [7, 51] to

expose prevailing themes. This approach is similar to qualitative coding and follows the same steps for qualitative analysis via coding as outlined by Auerback and Silverstein [3]. This is an appropriate method for semi-structured interviews as qualitative coding results in the possibility of applying the same code to different sections of the interview [31]. Moreover, affinity diagramming has had widespread use for qualitative data analysis over the last 50 years [48].

## 5 RESULTS

Participant's responses to interview questions and bi-weekly assessments provided insight into their experiences regarding AmbiTeam.

### 5.1 Quantitative Results

#### 5.1.1 Perceptions of Effort

We wanted to know if AmbiTeam affected researchers' perceptions of their collaborators' effort on a project (RQ2). To do this we conducted two quantitative tests. The first tested whether there was an effect of using AmbiTeam on the the researchers' *perception* of how much effort their collaborator was putting in. After establishing the normality of the user's ratings of their collaborators and self collected at the beginning and end of the study using Shapiro-Wilk normality test, we performed a paired samples T-test on the participant's ratings of themselves before using AmbiTeam (*self-baseline*, $\mu = 5.8$, $\sigma = 1.69$) and after the final week of using AmbiTeam (*self-post-use* $\mu = 5.5$, $\sigma = 2.07$). We found that there was no significant effect of AmbiTeam use on self-scores $t(9) = -0.45, p = .66$. In addition, we performed an independent T-test on the participant's ratings of their collaborators before using AmbiTeam (*collab-baseline*, $\mu = 5.94$, $\sigma = 2.13$) and after the final week of using AmbiTeam (*collab-post-use* $\mu = 6.33$, $\sigma = 1.53$). We found that there was no significant effect of AmbiTeam use on their scoring of collaborators $t(17) = -0.78, p = .45$.

The second tested whether there is a correlation between the average activity levels of their collaboration (as measured by AmbiTeam) and the researchers' *perception* of how much effort their collaborator was putting in. We performed a Pearson's product-moment correlation test on participant's average displayed activity (*activity*) and the change in personal ratings gathered during the bi-weekly assessment (*personal ratings*). We found no correlation ($r = 0.09$, $p > 0.05$) between *personal ratings* and *activity*. We also performed a Pearson's product-moment correlation test on *activity* and the change in ratings assigned to them by their collaborators during the bi-weekly assessment (*collaborator ratings*). We found a weak positive correlation ($r = 0.22$, $p = 0.011$) between *collaborator ratings* and the *activity*—as each participant's apparent activity increased, their collaborator's ratings of them increased.

In summary, using AmbiTeam clearly did not affect user's reported perception of their own effort; however, the conflicting results indicate that it may have affected the user's perceptions of their collaborator's effort. As a result of the contradictions in our quantitative results, we conducted follow-up semi-structured interviews to clarify AmbiTeam's affect on users' perceptions of their collaborator's effort.

### 5.2 Qualitative Results

#### 5.2.1 Interactions with the System

Most participants reported briefly looking at the visualization multiple times a day, often because the visualization was placed within their general field of view (although care was taken to ensure that the visualization did not obstruct the view of the participant's workstation). However, participants did not intentionally check the visualization for updates, indicating that the information generally stayed in the background.

> "It wasn't like I checked it intentionally several times a day. It was more of that I leaned back in the chair to

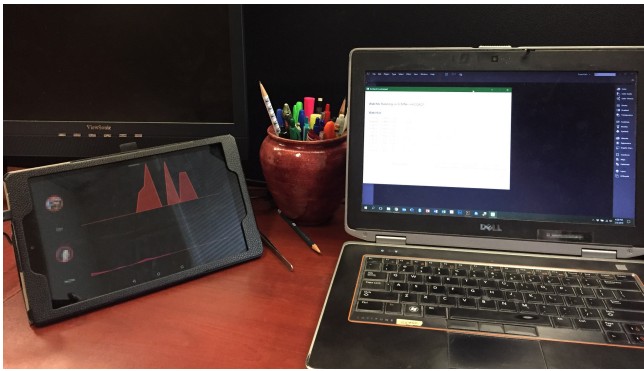

Figure 2: AmbiTeam's components shown in A1's workspace. The visualization was placed in a different location in the periphery of A1's attention during the study.

> think about something and while looking at other things in my desk. I would see it." C1

The information gleaned from the visualization was typically combined with information gathered during communications with collaborators. This information included knowledge about circumstances (e.g., job interviews, other papers and projects), project deadlines and updates, and each researcher's role in the project. In some instances the fact that collaborators were communicating at all was enough of an indication that those researchers were prioritizing the project. Participant B3, however, based their ratings solely on their communications with their collaborators because they did not trust AmbiTeam.

> "I couldn't place enough trust in the system yet to factor in positively or negatively into my perception of prioritization." B3

Most participants explicitly stated that using the system did not interrupt their workflow. This was partly due to the placement of the visualization within the user's workspace. Furthermore, the file tracking software was passive in nature such that once the user had selected their directories, no further action was needed. Participant C1 also remarked that the passive nature of the data collection resulted in more information than their usual workflow, because their usual workflow (GIT) relies on user to push information.

#### 5.2.2 Determining Engagement

To determine whether tracking file activity can give teammates a sense of their teammates efforts (RQ1), we asked open-ended questions during each bi-weekly assessment and conducted a follow-up interview at the end of the study. We found that participants felt AmbiTeam's monitoring method gave a measure of user engagement.

> "Tracking over time as you change it, it's simple so it does give you a measure of whether or not the person is engaged. Or not engaged. So I think it's a good measurement of that" C1

However, participants reported several activities that were not tracked by the system that were integral to their work. In general, these activities were related to collaboration, idea development, and management. Some of the suggested activities are likely fairly easy to take into account, such as tracking the number of files in a directory (e.g., a library of literature for a project), the size of files (e.g., as figures get made, manuscript and code gets written), written meeting minutes, and the number of times a program is run. Others

could be tracked by the existing software if the users change their behavior, such as making handwritten notes in a digital notebook as opposed to on physical pieces of paper.

However, many of the suggested activities (e.g., tracking emails, phone calls, internet searches, time spent on the top window of a computer) are difficult to take into account without invading privacy. Several participants stated that they wouldn't want personal data to be tracked unless it's somehow necessary for the team. Even then, participants requested caution when setting up AmbiTeam in order to prevent project-sensitive data from being tracked. For example, during the set up of group D, participants deliberately chose directories that contained metadata and statistics about the participants in their studies but did not contain identifiable data.

Finally, participants believed that for optimal use, the files and activities chosen for monitoring depend on the context of the user's work. They suggested that some metrics would be more suited to some roles than others. For example, since B4 was running user studies, the length of their files represents the amount of data collected and is more indicative of work than the number of files, which merely reflects the number of participants. Certain file types, such as those automatically created by ArcGIS [22] (a Geographic Information System Mapping Technology used by A1) and TensorFlow [53] models (a tool for building machine learning models used by B1) are automatically generated in bulk and don't necessarily indicate massive amounts of effort.

### 5.2.3 User Behaviors

To answer what affects the provision of team project activity information had on work behavior (RQ3), we asked open-ended questions during each bi-weekly assessment and conducted a follow-up interview at the end of the study. We found that on the whole, participants did not believe that using the system changed their collaborators' behaviors. However, many reported changing their own behaviors. In some cases, participants changed the way that their work was conducted to boost visibility and ensure that their collaborators knew that they were involved. For example, participant A2 described a time when they were creating a wiki for their project online. However, since AmbiTeam was unable to track the changes made to their online wiki, A2 wrote much of the text for the wiki on a text editor that saved changes to a file tracked by the system before uploading the text to the wiki. This ensured that their efforts to update the wiki appeared on the visualization. In addition to this, several participants mentioned saving their files more frequently so that their changes would register as activity and appear on the visualization.

Many participants reported that AmbiTeam made them feel more motivated to work on their projects. Sometimes this was due to participants noticing a lull in their own activity, which reminded them to work on the project. Motivation was also often attributed to seeing their collaborator's activity.

> *"Having a view of other people are working hard and then you don't want to be the last one. It's like a challenge."* D2

Participant A2 noted that the system as had a positive impact due to its effect on motivation and a desire to work effectively.

> *"Positive, because it helped motivate me to make the project a priority even though it's not the most fun thing to work on."* A2

### 5.2.4 Future Directions and Applications

All participants stated that they would be willing to use AmbiTeam, or a refined version of AmbiTeam, in the future for either professional or casual use. Several participants mentioned a desire to use the system in research collaborations to keep abreast of what their collaborators were up to. For example, participant C1 mentioned using the prior day's activity *"I could glance at as sort of like a morning statistics for yesterday."* Another use of the system would be for a project manager to balance the workload across researchers on a project, as described by participant B3 *"I probably would want to use it just to see how much work my each of my teammates is doing so that the load is balanced out evenly."*

Other participants reported that they would use AmbiTeam in a classroom setting both as a student working with group-mates that they don't know well or didn't pick and as professors managing class groups.

> *"I've had problems in the past ... they didn't do anything until the last week and even then in the last week, you know. I may have built the vast majority of it. They still get the same amount of credit."* C1.

Several participants also stated that they would use AmbiTeam for personal use. Participant A1 described not being interested in worrying about their collaborator's productivity, but was interested in using the system to take a *"long term perspective"* and revisit their own project-related activity. The goal would be to have a better understanding of the work that they had done in the past. In a similar vein, participant B2, a self-proclaimed *"data junky"* expressed an interest in using AmbiTeam to gain a deeper insight into their workflow. A1 also disclosed a belief that AmbiTeam could be useful for recent Ph.D. graduates who have transitioned from working solely on their dissertation to managing multiple projects and needing to have a better grasp of their priorities. Finally, A2 expressed an interest in using the system with a friend to stay motivated to work.

> *"In the same way that it's better to go to the gym with a friend because it motivates you because even on that one day when you really don't feel like going they'll go and then they'll help you get over that hump."* A2

Participants also expressed a desire to extend AmbiTeam to support additional tasks. For example, participants conveyed an interest in integrating AmbiTeam with task management systems, allowing users to connect the activity shown on the visualization with specific tasks and goals. Participant C2 also suggested incorporating a messaging system that would allow a user to contact a collaborator when they notice a lull in activity.

> *"[If] I made some changes that we needed to discuss that I could just look look at my collaborator and just tap ... saying hey, there's something that needs to be discussed."* C2

## 6 DISCUSSION

### 6.1 Motivational presence of others

Many of the participants reported feeling more motivated and productive while using AmbiTeam. These feelings can likely be attributed to the motivational presence of others [43]. Our participants' responses indicated they were aware of being watched by their teammates, which changed their behavior, as described by B1:

> *"Because I know we are being tracked, I want to make use of time to work efficiently."* B1

Researchers often use the presence of specific teammates in a shared space to guide their work [21]. Similarly, our participants also reported feeling motivated by seeing their collaborators work on the project, as stated by C2:

*"Every single time that happened I was like, oh he's working, I should probably work on it too."* C2

Unfortunately, these effects often dissipate once the participant no longer has a sense of the presence of their collaborators. Depending on the scientific questions that they seek to answer, researchers may spend time away from their desks where AmbiTeam is set up to perform fieldwork. More investigation is necessary to determine whether the increased motivation facilitated by the system is sustained when researchers are unable to access AmbiTeam.

## 6.2 Remote vs. Co-located Projects

Given the difficulties that researchers have maintaining awareness of their collaborators' work progress at remote locations without the ability to casually "look over their shoulder" [43], we expected that AmbiTeam would have a smaller effect on co-located participants' perceptions of their collaborators. In fact, participants from the co-located teams reported having an easier time determining their co-located collaborators' effort and reported having a smaller effect on their perception of their collaborator's priorities.

However, we found that AmbiTeam sometimes provided similar benefits to co-located participants as it did to remote participants. One co-located participant (C1) indicated that using AmbiTeam provided more information about their collaborator's effort than they got from their frequent communications with their collaborator — despite sitting next to each other. In this case, the information provided by AmbiTeam caused this co-located participant to change their expectations to take their collaborator's conflicting priorities into account. It's important to note that neither participant on Team C reported experiencing any negative effects from AmbiTeam's use. This finding indicates that AmbiTeam can be an effective tool even in co-located projects.

## 6.3 Privacy vs. Accurate Activity Tracking

During the post-study interviews, participants mentioned several activities that are part of their workflow that were not tracked by AmbiTeam during the study. However, tracking several of these activities would involve significant privacy violations, namely tracking in-person conversations, emails, and internet browsing history. This leads to the question of how to balance accurate activity tracking with maintaining user's privacy. It is possible that tracking additional, less-sensitive information (e.g., file length, degree to which a file has been changed) paired with customized tracking on a per-project and per-user basis may provide enough information that monitoring more-sensitive information like communications between collaborators is unnecessary. Further research is necessary to determine whether this is the case.

## 6.4 Future Work

One of the many dangers of remote work is loss of motivation. In co-located work, the presence of others has a large and important impact on teammates motivation [43]. We believe AmbiTeam was able to capture some of the motivational presence of others in remote work using an ambient display. In future work, we will explore other ways in which ambient displays can increase motivation.

Although tracking file activity allows us to gain some measure of effort, it does not encompass many important steps of work during the collaboration that could take plenty of time and effort to complete, but not change any file. For example, searching for resources online, brainstorming ideas and plans, etc. Future work can explore the use of different metrics for providing team awareness, such as the amount of progress on given tasks.

In addition, future work can also explore long term effects of systems like AmbiTeam to determine whether the immediate increase in productivity due to being watched decreases over long periods of time and see if tensions arise due to the limited display of team member's contributions. Furthermore, future work includes investigations

into the negative effect of tracking other people's working status all the time. For example, we would investigate whether AmbiTeam causes peer pressure or privacy concerns.

We evaluated AmbiTeam with collaborations of academic researchers who, while pursuing different research questions, had similar workflows. It is likely that all knowledge workers (workers who apply knowledge acquired through formal training to develop services and products [20]) can benefit from a system like AmbiTeam given that they generally have high amounts of screen time. However, it is less clear whether ambient displays work for all types of workers, including those whose jobs are very different from that of a knowledge worker (e.g., service work). In organizations with a clear hierarchy, does the role of the user affect the usefulness of AmbiTeam? Are there types of ambient data from a CEO that would motivate workers? For this reason, future work includes exploring the use of AmbiTeam in a variety of contexts of work.

It is also unclear how well ambient displays work for providing activity information in large teams. Our assessment of AmbiTeam was with small teams of 2-4 people. How well will a system like AmbiTeam work for an entire organization? Given that organizations are frequently divided into smaller teams, is there even a need for systems like AmbiTeam to work with large collaborations?

Many collaborations are highly temporally dispersed, sometimes operating across extreme time zone differences. In these situations, such as with a 12 hour time zone difference, people aren't working at the same time. Can we still effectively summarize progress from their work? Is the provision of activity information about a coworker who is not working at the same time still motivating?

Finally, future work can harness the iterative design process to refine and develop AmbiTeam to meet the expectations of users. Further evaluation can determine the effectiveness of AmbiTeam in meeting the needs expressed by our users in this study.

## 7 Conclusion

In this paper, we described and evaluated a system meant to assist researchers experiencing the problem of perceived prioritization. We found that, despite shortcomings with regards to activity tracking, AmbiTeam had some effect on user's perceptions of their collaborators' effort as well as their motivation to work on their collaborative project. This work has implications for creating effective awareness-based technology for supporting collaborative work, particularly the recommendation that future awareness systems consider (a) using file activity to measure effort and (b) implementing ambient displays that do not interrupt the user's workflow. Furthermore, this paper contributes new knowledge regarding the use of computer graphics to influence perceptions of effort in remote teams.

## ACKNOWLEDGMENTS

We would like to thank Halle Parris for her assistance in developing AmbiTeam's visualization.

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
