# OpenReview forum: "AmbiTeam: Providing Team Awareness Through Ambient Displays"
_graphicsinterface.org/Graphics_Interface/2021/Conference/Second_Cycle — GI 2021_

### Official Review · Reviewer_Ca7b · 2021-04-26
**Review of "AmbiTeam: Providing Team Awareness Through Ambient Displays"**

**Rating:** 5
**Confidence:** 3

**Review:**

The paper presents the AmbiTeam, an ambient display that shows activity related to the files of a team project, to help collaborations preserve a sense of the team’s involvement while working remotely. A 10-participant user study was conducted to determine whether AmbiTeam facilitates team awareness. The study results show that AmbiTeam had some effect on user’s perceptions of their collaborators’ effort as well as their motivation to work on their collaborative project.


Overall, the paper is well-motivated, well-written, and easy to follow. The questions that the paper is trying to answer are interesting and clear. The study seems solid, and the statistical analysis is thorough and complete. However, there are still some issues in this paper that need to be addressed to make it a meaningful paper. Below is a list of weaknesses.


**Activity of Files**

AmbiTeam tracks and visualizes file changes in user-specific project directories to indicate how much effort or work a collaborator has put in on the project. I am not sure if file changes are good indicators to represent work effort. There are many other tasks during the collaboration that could take plenty of time and effort to complete, but not change any file. For example, searching for resources online, brainstorming ideas and plans, etc. Authors might need to justify why file changes are representative and can stand for the current workload/ working status.



**Comparison with Existing Collaborative Tools**

In the PRIOR WORK section, the authors discussed many existing collaborative tools, but it is hard to understand the difference between the existing tools versus AmbiTeam, which makes the paper contribution a bit vague. It seems to me that some collaborative tools have already provided “effort visualization”. Thus, it is essential to elaborate on the comparison and what is new for AmbiTeam.



**Ambient Display**

Authors argued that AmbiTeam on an iPad nearby serves as an ambient display and does not interrupt/influence users’ primary workflow. Although it seems true for at least one participant [C1], I am not sure if this is always valid for all of the users under different conditions. Authors might need to provide more discussions and justifications regarding ambient display – how can we ensure AmbiTeam is “ambient”? Why AmbiTeam is “ambient”? To me, it might be disturbing if an iPad nearby keeps showing me “moving” line graphs.

**Negative Effect**

Sometimes [or for most of the time], it is good to know the collaborator’s work progress. However, I also hope authors can discuss the negative effect of tracking other people’s working status all the time. Would it cause peer pressure or privacy concerns?

**Summary**

In a word, although the paper has many merits, I think it still needs another round of revision to make it a good addition to GI.

---

### Official Review · Reviewer_aJxf · 2021-05-02
**Simple system and study, yet non-trivial findings.**

**Rating:** 7
**Confidence:** 4

**Review:**

The paper "AmbiTeam: Providing Team Awareness Through Ambient Displays" presents an ambient display, AmbiTeam, that can visualize activity of a work team based on edits to selected files and folders. AmbiTeam is deployed with 10 scientists in three work teams, and its effect is evaluated through qualitative data analysis of semi-structured interviews with the participants. The participants did not change their perception of their own effort, but it did affect the perception of collaborators' effort. Furthermore, the study demonstrated that the participants would adapt their behavior to perform actions that would trigger the activity tracker (e.g., produce content for a wiki in text files to have the system track the work).

Overall, I enjoyed reading the paper. It is well written, well-motivated, and covers related work nicely.
AmbiTeam is a simple system, but the findings are interesting and non-trivial. There are obviously limitations with only tracking file system activity, but the authors also acknowledge this. I miss details on how an activity is configured and how the boundaries of a project/activity are negotiated between colleagues.
For future work in going beyond file-based activity tracking, I would encourage the authors to survey tools for activity tracking (e.g., I know of https://timingapp.com/ for Mac). There are tools that very detailed can track activity, but to use them similarly, like AmbiTeam would require mechanisms to control what is shared and to what detail with collaborators.

Overall, I recommend accepting the paper for Graphics Interface 2021.

---

### Official Review · Reviewer_y2ti · 2021-05-04
**AmbiTeam: Providing Team Awareness Through Ambient Displays**

**Rating:** 5
**Confidence:** 4

**Review:**

The paper presents the development and preliminary evaluations of the AmbiTeam – which is an ambient display in the workspace to help researchers visualize their contributions (number of changes in the file) in the project by using the principle of “motivational presence of others”. For evaluation purposes, the authors conducted a 10-participant user study to determine whether the AmbiTeam had some impact on the researchers’ motivations and contribution to the project. The study results show that AmbiTeam has some potential to create that impact on the collaborator’s motivation. Overall, the paper is well structured and well written, but it can be improved.

Here is the review of individual sections and the things that can be improved in these sections.

Introduction:
The introduction section is very brief. The author can cite more work regarding the shift of use in the way collaborators use tools before and during the pandemic to stress the importance of developing and improving these tools. The author clearly sets the ground for the research gap and where AmbiTeam fits in and how the AmbiTeam meets up to that, this is good. Also, the authors clearly mentioned the contributions of this paper.

Related Work:
In the previous work, the author has mentioned multiple systems previously designed in the context of ambient displays. As a reader, it makes me really curious to see the section in the related work which specifies the summary from this related work and presents how AmbiTeam is different from them. Since the paper is below the page limit, adding a new section will not be an issue here. Also, the necessary explanation for choosing “changes in the file” as a factor to visualize the effort is missing in the related work. The changes in a project can be qualitative as well as quantitative in nature. The best way to determine this is to set an impact score to each of the changes and visualize the same on the AmbiTeam dashboard. Authors must specify how the “changes in the file” were selected as the factor for this research.

System Design:
Authors need to justify their point of using “file change activity” as a measure for the changes in various phases of SDLC. This can be improved by citing previous research in a similar or another domain. Also, ambient design by definition should not interrupt users’ primary focus. Looking at figure 1, it seems like you have to make a focus shift and see that graph specifically on the iPad to make meaning out of it. It can actually be difficult for some participants.

Methodology:
There are some shortcomings in the methodology section. The authors conducted a pre-test with quantitative scales, but the post-test study was just a semi-structured interview. Both the tools are not comparable. Authors should have considered using the same scales in the post-test and then comparing the significant differences in the results and maybe try to use the interview comments to support these results. Based on the current methodology, it is very difficult to say if the results can be relied upon. Also, if there were multiple assessments bi-weekly, as a reader I am curious to see differences in scores over time.


Results:
One of the comments that the participant can up with was this: “I couldn’t place enough trust in the system yet to factor in positively or negatively into my perception of prioritization.” B3.
This clearly shows that maybe the AmbiTeam interface needs a drill down into the graphs for teammates so that there can be enough details of what tasks are performed to achieve that kind of graph. This kind of interface may be helpful for the researchers to advocate transparency of the system and it will be useful to get the trust of the users. The author mentions that most of the participants felt that it was not an interruption to the workflow. It will be better if they can specify the numbers instead of using the terms like most and some. Participants also mentioned some changes in terms of additional features that can be added to the AmbiTeam. For example, chat application, task management, goal creation, and management.

Discussion:
It seems like the author has considered the paper “Bridging Distance: Empirical studies of the distributed team” as their base paper and most of the references are linked here. But I was really expecting to see a strong discussion of this work where the author compares their results with multiple other studies and talks about whether the results are inline or different. The discussion section needs revision to include all of this.

Future Work and Conclusion:
The authors stated the limitations of their study in this section as well as the methods they will use to accomplish them. Apart from the limitations mentioned – The authors must also keep in mind to use the iterative design process for future studies. In this manner, it can really be tracked what changes are expected by the users and how they are being delivered. The conclusion section is very brief. The author must think about the contributions to the GI world and phrase a sentence or two to improve this section.

---

### Meta-Review · Area_Chair_Vm4k · 2021-05-07

**Recommendation:** Accept
**Confidence:** 4

**Metareview:**

The reviewers agreed that the paper is well-written, well-motivated, easy to follow, and relevant to GI.

However, they also highlighted some weaknesses of the paper in its current form that undermine the contribution. Below, I summarize the key ones but encourage authors to go through the reviews for other issues.

-	It is unclear how is the current paper is different from already existing work. A comparison of current findings with existing similar works will be necessary. The unclarity between this current work and other existing work makes the contribution of the paper difficult to articulate.
-	Choice of visualization factors, tracking only the file system is a problem. It may not be a good indicator of work effort.
-	Weaknesses in the methodology especially regarding the pre- and post-study methods and comparison factors.
-	Lack details that would help in understanding the methodologies such as how activities are configured and the boundaries,
-	Needs to provide more discussions about the Ambient display to demonstrate that it’s really ambient and not distractive.
-	Need for discussion on the likely negative effect of tracking other people’s work such as peer pressure

---

### Decision · Program_Chairs · 2021-05-08

Accept